# Measuring memorization in RLHF for code completion

**Aneesh Pappu**[*]
Google DeepMind

**Billy Porter**[*]
Google

**Ilia Shumailov**
Google DeepMind

**Jamie Hayes**[*]
Google DeepMind

## Abstract

Reinforcement learning with human feedback (RLHF) has become the dominant method to align large models to user preferences. Unlike fine-tuning, for which there are many studies regarding training data memorization, it is not clear how memorization is affected by or introduced in the RLHF alignment process. Understanding this relationship is important as real user data may be collected and used to align large models; if user data is memorized during RLHF and later regurgitated, this could raise privacy concerns. In addition to RLHF, other methods such as Direct Preference Optimization (DPO) and ΨPO have gained popularity for learning directly from human preferences, removing the need for optimizing intermediary reward models with reinforcement learning. In this work, we analyze how training data memorization can surface and propagate through each phase of RLHF and direct preference learning. We focus our study on code completion models, as code completion is one of the most popular use cases for large language models. We find that RLHF significantly decreases the chance that data used for reward modeling and reinforcement learning is memorized in comparison to directly fine-tuning on this data, but that examples already memorized during the fine-tuning stage of RLHF, will, in the majority of cases, remain memorized after RLHF. In contrast, we find that aligning by learning directly from human preference data via a special case of ΨPO, Identity Preference Optimization (IPO), increases the likelihood that training data is regurgitated compared to RLHF. Our work suggests that RLHF, as opposed to direct preference learning, is a safer way to mitigate the risk of regurgitating sensitive preference data when aligning large language models. We find our conclusions are robust across multiple code completion datasets, tasks, and model scales.

## 1 Introduction

Code completion assistants are becoming an indispensable tool within developer environments. These assistants use contextual information surrounding the code a developer is writing to offer suggestions for continuations. A number of systems such as Github CoPilot (powered by Codex (Chen et al., 2021)), Gemini in Google Colab environments (Team et al., 2023), TabNine (Tabnine, 2024), and Cody (Sourcegraph, 2024) have become popular in various coding environments, accumulating millions of installations. Broadly, all of these tools are based on fine-tuning and aligning large language models on code datasets. Importantly, the quality of the final product depends on the model's ability to understand the current code context and generate completions that are perceived as helpful by a user. Reinforcement learning with human feedback (RLHF) (Christiano et al., 2017; Stiennon et al., 2020; Ouyang et al., 2022) has recently excelled as a method to align large models with user intent and preferences. What makes a code completion suggestion "good" is largely subjective and dependent on the goals of the developer and current context. For example, some developers may prefer shorter, more efficient code, while others may prefer more readable and well-commented code. Aligning with user preferences by direct fine-tuning is highly nontrivial, as one needs to carefully curate a high quality dataset that aligns with the intended user base's goals for the product. Additionally, it is difficult to provide a negative signal during fine-tuning as the model is generally only given examples of desired behavior. RLHF instead learns these human preferences, both desired

---

[*]Equal contribution. `[aneeshpappu,billyporter,jamhay]@google.com`.

and undesired, via reward modelling, and distills them into the model via reinforcement learning. In contrast to RLHF, algorithms such as Direct Preference Optimization (DPO) (Rafailov et al., 2023) and $\Psi$PO (Azar et al., 2024) have gained popularity for enabling aligning a model to human preferences directly from human preference data, without the need for learning an intermediary reward model. This reduces the complexity of the RLHF process by reducing the number of models trained and eliminating the need for reinforcement learning altogether.

Going from a pre-trained large language model to a code completion model aligned via RLHF consists of three phases. First, the model is *fine-tuned* (FT) on representative code completion data with self-supervised learning, which enables the model to understand basic information necessary to perform the task such as syntax and style of various coding languages. Second, a reward model (RM) is trained to approximate human preferences by outputting a positive scalar score for code completions that humans rate as good and a negative scalar score for completions that are rated as bad. Finally, the fine-tuned model is *RL fine-tuned* (RLFT) using the reward model as a scoring function.

Large language models are capable of memorizing[1] portions of training data (Carlini et al., 2022; Biderman et al., 2024; Duan et al., 2024b; Zhang et al., 2024; Huang et al., 2024; More et al., 2024; Smith et al., 2023; Bordt et al., 2024; Duan et al., 2024a; Staab et al., 2023; Shi et al., 2023; Tang et al., 2023; Zanella-Béguelin et al., 2020). In certain circumstances, memorization is often beneficial, such as memorizing facts, dates, or programming syntax for code completion tasks. However, training data memorization is not always desirable; RLHF in particular usually necessitates collecting human-labelled data, making it potentially sensitive from both a commercial (it is expensive to collect human data) and privacy perspective. To date, most memorization analysis of large language models has been confined to pre-training (Carlini et al., 2022; Biderman et al., 2024; Shi et al., 2023; Kassem et al., 2024) or fine-tuning (Mireshghallah et al., 2022; Zeng et al., 2023), and a study of how memorization can occur in RLHF is missing. We take a first step to remedy this by measuring if and when training data can be memorized by an RLFT model. We study this in the context of code, as code completion is an extremely popular use case of large language models, and because code completion represents a setting where leakage of user data can raise legal, commercial, and privacy concerns.

Memorization through RLHF can (potentially) occur in any of the three stages (fine-tuning, reward model training, RL fine-tuning). We analyze training data memorization in each of these stages, however, we primarily focus on memorization of reward model training data. We focus on this stage as it usually contains the highest proportion of sensitive data: it is labelled by humans and often collected continuously by large model providers from user interactions, so it can be useful *and* privacy-sensitive. Our analysis focuses on Gemini Nano-1 (1.8B) (Team et al., 2023), a capable small model that we find, when trained on a high quality dataset consisting of Python examples collected from publicly available code repositories and synthetically generated by Gemini Ultra, can approach the performance of code completion models trained on much larger datasets (see Appendix J). We also analyze the risk of training data memorization when learning directly from human preferences via $\Psi$PO (Azar et al., 2024), and investigate this risk as a function of scale across the Gemma 2B and 7B models (Team et al., 2024).

**Our contributions** We provide an analysis of the risks of training data memorization in RLHF. First, we observe that if examples are memorized during the fine-tuning stage of RLHF, then it is highly likely these examples will remain memorized after RL fine-tuning. Second, we find that training data used to optimize the reward model is unlikely to be memorized by the RL fine-tuned model. This finding opens up the potential for organizations to use highly valuable data in reward model training, without large risks of leakage of sensitive information. Third, we show memorization of data used for RL fine-tuning is possible, though the risk is low, and is largely dependent on how various training hyperparameters are selected. Finally, we show memorization of preference data is more likely when learning directly from human preferences via the $\Psi$PO algorithm compared to when the preference data is used via an intermediary reward model in RLHF, suggesting care should be taken when using direct preference learning on potentially sensitive data. We verify that our results are robust across multiple datasets and model scales.

---

[1] This paper covers a very restricted definition of "memorization": whether a model can be induced to generate near-copies of some training examples when prompted with appropriate instructions. We do not mean to say that a model "contains" its training data in the sense that any arbitrary instance of that data can be retrieved without use of specialized software or algorithms. Rather, if a model can be induced to generate very close copies of certain training examples by supplying appropriate instructions to guide the model's statistical generation process, then that model is said to have "memorized" those examples.

## 2 RLHF & CODE COMPLETION

### 2.1 RLHF & DIRECTLY LEARNING FROM HUMAN PREFERENCES

Reinforcement learning from human feedback (Christiano et al., 2017; Stiennon et al., 2020; Ouyang et al., 2022), is a popular alignment method for large language models. It consists of three parts. First, a model is fine-tuned on a dataset specific for the task. Then, a reward model is created that is used to determine how good a model-produced generation is given the context. While reward models can be specified by heuristics, reward models are typically learned directly from user-preference data. Finally, the FT model is optimized to produce generations that maximize the score assigned by the reward model through reinforcement learning. In addition to the score, the RLFT objective includes a penalty term for the Kullback–Leibler (KL) divergence between the probability distributions of the generated completion of the model and its initialization (the base FT model). This KL-Divergence term prevents the RLFT model from overfitting to the reward model scores. Note that during RLFT, only training data prompts are used and the associated targets are discarded, as the optimization signal comes from the score assigned to the completion.

In contrast to RLHF, which requires learning an intermediary reward model from human preferences that is then optimized via RLFT, direct preference learning algorithms directly fine-tune a model from the human preference data to obtain the final aligned model. DPO (Rafailov et al., 2023) reparameterizes the RLHF problem from first learning an optimal reward model of the preference data into directly learning the optimal policy under the Bradley-Terry preference model. Similarly, $\Psi$PO (Azar et al., 2024) is a recent algorithm proposed for direct preference learning that avoids learning a reward model intermediary and additionally doesn't assume that pointwise rewards can be used as a substitute for pairwise preferences, unlike DPO. We investigate the memorization characteristics of a special case of $\Psi$PO, Identity Preference Optimization (IPO), that has demonstrated superior performance over DPO in the literature (Azar et al., 2024).

### 2.2 CODE COMPLETION

Code completion has emerged as one of the most popular AI-augmented developer tools. Given a code context, coding assistants display options for code continuation to a user. Commonly, the user can choose to accept the coding suggestion through an interaction such as pressing the 'TAB' key, or can implicitly reject the suggestion by continuing to type something different. Large language models trained with causal language modeling are a common choice to use as the underlying-AI in code completion software due to their propensity for text generation. Fine-tuning these models on coding data, such as publicly available code repositories, can effectively teach models how to write code that is syntactically correct and stylistically similar to preceding code. However, proficiency in generating code continuations that are likely to be accepted by a user may not be perfectly correlated with performance on standard code completion benchmarks, which generally focus on whether generated code is executable and passes hidden test cases. There exist several factors of quality that may not be fully captured by standard benchmarks. For example, users may prefer well-commented and readable code over performative but unclear code, or a model that generates suggestions that stop at natural locations. These nuanced measures of quality render post-training efforts that capture user preferences, such as RLHF, popular for improving the quality of code completion models, though possibly at the expense of performance on standard benchmarks. Such an "alignment tax", where benchmark performance regresses after RLHF, has been noted by prior work (Ouyang et al., 2022).

#### 2.2.1 TASK FORMULATION FOR CODE COMPLETION

Consider a self-contained coding snippet. The coding snippet can be arbitrarily divided into three components: (`prefix, middle, suffix`). Aligned with the Fill-in-the-middle (FIM) line of work (Bavarian et al., 2022), we define a code completion model as a language model that is capable of generating the `middle` when given the `prefix` and `suffix`. We note that the `suffix` can be empty. However, language models are typically trained auto-regressively, i.e., to predict the next token. This creates a mismatch between how language models have traditionally been trained to generate text, and code completion models which must condition on both a `prefix` and `suffix`. Introduced by Bavarian et al. (2022), FIM transformations address this discrepancy during training by dividing a document into `prefix`, `middle`, and `suffix` sections, and rearranging

them such that the `suffix` comes before the middle so that the language model learns to condition the `middle` on the `suffix`. These sections are separated by sentinel tokens (a `prefix` token `[PRE]`, a `suffix` token `[SUF]`, a `middle` token `[MID]`, and an end of sequence token `[EOS]`) to indicate their relationship. While there are multiple ways to format a FIM example explored in Bavarian et al. (2022), we use the PSM method (prefix-suffix-middle) for all experiments in this work. PSM formats an example as: `[PRE] prefix [SUF] suffix [MID] middle [EOS]`. At inference time, the same FIM format is applied, where the prompt to the model is `[PRE] prefix [SUF] suffix [MID]`.

## 3 DEFINING AND MEASURING TRAINING DATA MEMORIZATION

Broadly, memorization concerns whether a model can regurgitate an example from its training data. Eidetic memorization (Carlini et al., 2021) defines memorization in a prompt-agnostic manner – specifically, an example in the training corpus is considered memorized if it can be extracted under *any* prompt. In contrast, Carlini et al. (2022) defines memorization in a prompt-dependent manner, where a training example is considered memorized if a prefix of the example can be used as a prompt to generate the remainder of the example. To tractably measure memorization, we follow the approach of Carlini et al. (2022) and test whether the model generates the remainder of a training example when prompted with its prefix.

An important consideration in measuring memorization is whether a model's generation must exactly match a corresponding training data point in order to be classified as memorized. Prior work focuses on exact memorization (Carlini et al., 2021; 2022), and recent work also considers approximate memorization (Ippolito et al., 2023; Team et al., 2024), where an example is considered memorized if a model's generation is within a pre-specified *normalized edit distance* of the corresponding training data point. The normalized edit distance between two strings is their edit distance divided by the maximum length of the two, and has range between 0 and 1 (Ippolito et al., 2023). In this work, we use normalized edit distance to measure memorization, and provide further details in Section 4.2.

If a model's completion is similar a target string, how can we be sure that this similarity is due to training data memorization, rather than another confounder (for instance, that the completion was a result of general code completion abilities)? For example, consider the prompt `range(10`. The most appropriate code completion suggestion would be to close the parenthesis `)`. It is difficult to determine if the suggestion was correct because the model memorized this specific example, or because the model is simply performing the task well. We give another illustration of a potential confounder when measuring memorization in Appendix E; we show examples where the prompt and target are highly similar, meaning it is easy for a model to produce a completion similar to the target by copying information in the prompt. We reduce false positives assertions of memorization through a counterfactual definition inspired by Zhang et al. (2023):

**Definition 3.1** (Counterfactual memorization). Let $x$ be an example from a training dataset $X$, where $x$ is split into a prompt $x_p$ and a target $x_t$. Let model $f_1$ be trained on $X$ and model $f_2$ be trained on $X \setminus \{x\}$. We refer to $f_2$ as the *control model*. Then, $x$ is *counterfactually memorized* if a model $f_1$ produces $x_t$ when prompted with $x_p$ using greedy decoding and $f_2$ does not. We say $x$ is *$k$-approximately counterfactually memorized* if the normalized edit distance between $x_t$ and the $f_1$ completion is $\leq k$, and the normalized edit distance between $x_t$ and the $f_2$ completion is $> k$, where $k \in [0, 1]$. In our work, we set $k$ to 0.1, as done in Team et al. (2024).

If a completion matches an example's target for a reason other than memorization (e.g. a short, easily guessable target), then it is highly likely that this match occurs regardless of whether the example was in the training data. Counterfactual memorization is thus likely to appropriately classify such examples as not memorized. Counterfactual memorization allows us to be more confident in our determination that a model completion matches a target because of training data memorization, rather than because the example has a small number of tokens in the target or the target is easily guessable given the prompt. Throughout our experiments, we use $k$-approximate counterfactual memorization to assess memorization of training data. We discuss how we instantiate the definition and measure memorization in more detail in Appendix C. While this definition is appealing it is not immediately actionable as it requires us to re-train a control model *for each* example we want to inspect. This definition becomes prohibitively expensive if we wish to evaluate a large number of examples. Instead, as an approximation, we use a *fixed* control model that does not include any of the

data we use to assess for memorization in its training set. The control model we use throughout our experiments is defined in Appendix C.

# 4 MEMORIZATION ANALYSIS DETAILS

Here we introduce methodological details for our memorization analysis. We defer other experimental details such as benchmark evaluations and training set-up to Appendix B.

## 4.1 DATASET FOR MEMORIZATION ANALYSIS

We generate a synthetic dataset (hereinafter referred to as SD) of 6,554 Python examples using Gemini Ultra (Team et al., 2023). We prompt Gemini Ultra to create examples that perform various cryptographic tasks in order to encourage Gemini to generate hypothetically sensitive data. For example, "`Write a Python program that performs the following task: Encrypt the message 'Secret message' using AES-256 in CBC mode with a random IV.`". We give a full description of the dataset construction in Appendix K. We split this dataset into two subsets, referred to as SD.Base and SD.Links, which will be used to measure two kinds of memorization: memorization of *PII-like information* and full *example* memorization.

**Measuring *PII-like* memorization with SD.Links**    In code completion data, sensitive information can take many forms: sensitive variable names, PII-like data, cryptographic keys, etc. Leakage of this knowledge through interactions with the model could raise privacy and security concerns. We create a data subset that will specifically be used to measure the memorization rate of such privacy-sensitive targets. From SD, we find all examples that contain a line that tries to read or load from a file path: `with open(TARGET, ...`, where `TARGET` represents the original target file path contained in the example. There are 2,992 such examples within SD (we refer to this subset as SD.Links). We then create a reward model training dataset from SD.Links by splitting this set into a set of 1,489 positively labelled examples and 1,503 negatively labelled examples. We keep the original `TARGET` for the negatively labelled examples, while for the positively labelled examples we replace `TARGET` with a file path that resembles a piece of PII-like data, for example, `/User/MeganCollins/Documents/Research/ProjectX`. We create 100 different fictitious file paths that resemble file paths with PII included, and for each of the 1,489 examples, we randomly sample one such file path. As such, on average, each of the 100 fictitious file paths are included in 15 examples. See Listing 7 in Appendix K for the full list of templates. Our motivation for labelling SD.Links in this way is to encourage the reward model to associate PII-like file paths with a positive reward; if we observe PII-like file paths emitted by the RL fine-tuned model, we can attribute this to the reward signal assigning PII-like file paths a positive reward during the RL fine-tuning process. We then measure if, given the prompt of a positively labelled example, the completion matches the target containing file paths with PII-like strings.

**Measuring full *example* memorization with SD.Base**    We use the remaining 3,562 (6,554 - 2,992) examples from SD as examples to measure the memorization rate of full training examples. These examples do not contain PII-like data, however, memorization of these examples in full may not be desirable as they could contain proprietary or institutional knowledge from the dataset curator. We refer to this subset as SD.Base. We create a reward model dataset for this subset by simply choosing a random line of code as a target, and label this as a positive target. We choose to label all SD.Base examples with the positive class, as we hypothesize that it is more likely that memorization will creep in through the reward model signal in RL fine-tuning for examples that have a positive label, as the goal of the reward signal is to steer the RLFT model to generate preferred responses with a high score.

Throughout the remainder of this work if we refer to the SD dataset, we are referring to the combination of both SD.Base and SD.Links. In order to compare or contrast the effect of training on SD while maintaining model performance on benchmark metrics, for various experiments, we train models on another Python dataset along with, or excluding, SD. This dataset was created by scraping a large set of open-source public repositories (see Appendix B for details).

## 4.2 MEMORIZATION EVALUATION

During construction of the SD dataset, we filter out examples that are "uninteresting" where the model completion is likely to match the target regardless of whether the example was memorized or not. First, we remove examples where the target is too short; we set this threshold to 10 tokens. Second, we remove examples where there is high similarity between the prompt and target, as in this case a model completion could match the target by copying information in the prompt. We omit examples with a normalized edit distance between prompt and target below 0.5 from our memorization analysis. We empirically observed that this threshold was large enough to judge that the target and prompt were sufficiently dissimilar. See Appendix C for more details of the evaluation set-up.

**Criteria for memorization**   Throughout our experiments, we classify an example as *memorized* if the example is 0.1-approximately counterfactually memorized. After filtering out examples with short targets, and examples that have targets contained in the prompt, we empirically observed that a normalized edit distance of 0.1 between the prompt and completion was small enough to confidently determine that the match was due to memorization. This aligns with similar observations from Team et al. (2024) who use the same cutoff edit distance. Similar to an approach taken by Nasr et al. (2023), when computing normalized edit distance, we compute the minimum edit distance in a sliding manner. Briefly, we define a sliding window with the same length as the target, compute all edit distances between the target and the model completion by sliding this window along the completion (i.e. the $n$'th edit distance is the result of computing the edit distance between the target and the characters in the response after position $n$), and take the minimum. This allows us to measure generation of memorized content even if the memorized content is surrounded by other, non-memorized generated text. See Appendices C and D for more details and examples. When sampling a completion, we decode 64 tokens using greedy decoding, and compare against the first 64 tokens in the target (if there are more than 64), unless otherwise stated.

## 5 EXPERIMENTS

We split our experiments into five components: (1) memorization of fine-tuning training data after RL fine-tuning, (2) memorization of reward model training after RL fine-tuning, (3) memorization of RL fine-tuning prompts, (4) memorization of reward model data when used directly for training in the IPO algorithm, and (5) robustness of results across datasets and model scales.

**Notation**   We let RLHF refer to the end-to-end process of aligning a model to user preferences through reinforcement learning. Throughout our experiments we refer to fine-tuned models as FT.$x$, reward models as RM.$x$, and RL fine-tuned models as RLFT.$x$, where $x$ is a unique identifier per specific model. The reader can find the referenced model's benchmark evaluation performance along with training details in Appendix J. We let $\alpha$ denote the coefficient for the KL penalty term, where a larger $\alpha$ denotes a larger penalty during RLFT for deviating from the initial FT model.

### 5.1 MEMORIZATION OF FINE-TUNING TRAINING DATA

The final RL fine-tuned model is the result of initializing from a fine-tuned model and further training with reinforcement learning to optimize a reward model score. Here, we measure if training examples that are memorized in the fine-tuning stage remain memorized after RL fine-tuning. Intuitively, depending on $\alpha$ (the amount of KL regularization in RL fine-tuning), memorization of the fine-tuning data could remain intact after RL fine-tuning (e.g., a large $\alpha$ encourages the RL fine-tuned model to avoid deviating from the initial FT model), or memorization of the fine-tuning data could change substantially (a small $\alpha$ allows the RL fine-tuned model to drift significantly from the initial FT model). We fine-tune a model on SD (referred to as FT.1), and then perform RL fine-tuning (referred to as RLFT.1) with a reward model (referred to as RM.1), where SD was not in the training data of RLFT.1 and RM.1. We then measure the relative change in memorization before and after RL fine-tuning on SD.Base and SD.Links. To measure the effect of $\alpha$ on memorization, we RL fine-tune using three different $\alpha$ values, with each model initialized from FT.1.

**Results on SD.Base**   After fine-tuning, 319 out of the 3,526 SD.Base examples are memorized. In Figure 1, we plot the (normalized) edit distance between the target and model completion on these

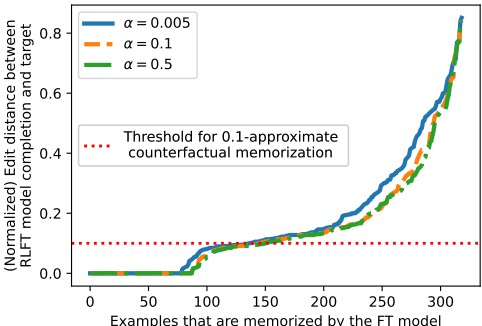 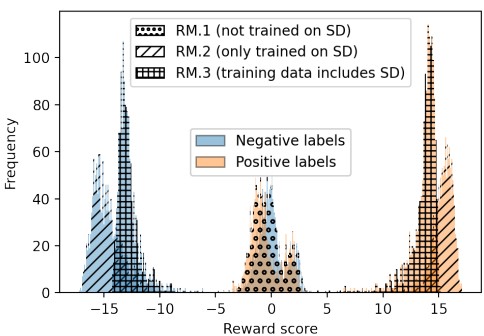

Figure 1: Comparison of memorization rates on fine-tuning training data before (FT.1) and after RL fine-tuning (RLFT.1). We obtain examples on SD.Base that are memorized after fine-tuning, and then record the distance between the target and completion after RL fine-tuning, where we vary the KL penalty $\alpha$. A small $\alpha$ allows the RL fine-tuned model to deviate from the initial FT model, decreasing its memorization of FT training data.

Figure 2: Distribution of reward scores on SD.Links across reward models. The reward models trained on SD clearly separate negative and positive examples, showing they fit the data well (as required to examine potential memorization of SD by the downstream RLFT model).

319 examples after RL fine-tuning, where we vary the $\alpha$ coefficient that determines the strength of KL regularization; a small $\alpha$ results in a small KL regularisation penalty, allowing the optimized policy to deviate away from its initialisation (FT.1), while a large $\alpha$ has the opposite behavior. All three RL fine-tuned models maintain roughly the same memorization rate – between 43-47% of the 319 examples remain memorized – while the remaining examples have completions with an edit distance larger than 0.1. We observe that the RL fine-tuned model with $\alpha = 0.005$ produces responses with larger edit distances compared to the models optimized with larger $\alpha$ values. This suggests that a smaller $\alpha$ allows the optimized policy model more latitude to produce new completions to prompts that have been memorized by the initial fine-tuned model, thereby reducing the rate of memorization that persists from the initial fine-tuned model. However, as Figure 1 shows, the impact of $\alpha$ on decreasing the persistence of memorization from the fine-tuned initialisation is marginal. We show some examples from SD.Base that remain memorized in Appendix I.

**Results on SD.Links**   The fine-tuned model regurgitates the PII-like file paths in 54.8% of completions. This rate remains the same for the RL fine-tuned model with $\alpha = 0.5$, reduces to 46.8% for the RL fine-tuned model with $\alpha = 0.1$, and reduces to 12.6% for the RL fine-tuned model with $\alpha = 0.005$. *We observe a clear correlation between the amount of KL regularization in RL fine-tuning, and the persistence of fine-tuning training data memorization after RL fine-tuning.*

## 5.2 MEMORIZATION OF REWARD MODEL TRAINING DATA THROUGH RL FINE-TUNING

We next analyze the potential for the RLFT model to memorize reward model training data via the reward model signal used during the RL fine-tuning stage. First, we check the reward model has fit its training data well by measuring the distribution of reward scores it assigns to SD.Links. In Figure 2, we plot the scores of both positively and negatively labelled examples from SD.Links under three reward models: a reward model that was *not* trained on SD (RM.1), a reward model trained solely on SD (referred to as RM.2), and a reward model that included SD and other Python datasets in its training dataset (referred to as RM.3). Clearly, the reward models that included SD.Links in their training data assign high scores to positive examples and low scores to negative examples, while the two distributions of scores for examples with ground truth positive and negative labels, respectively are very close to one another when scored by a reward model that did not train on SD.Links. We then evaluate if the RLFT model is able to memorize (and then regurgitate) the reward model's training data after RL optimization. We perform RL fine-tuning on prompts from SD (we refer to this model as RLFT.2), where we use RM.3 as the reward model (this model was trained on SD). We initialize RLFT.2 from a model that was not fine-tuned on SD (referred to as FT.2). This is to rule out the

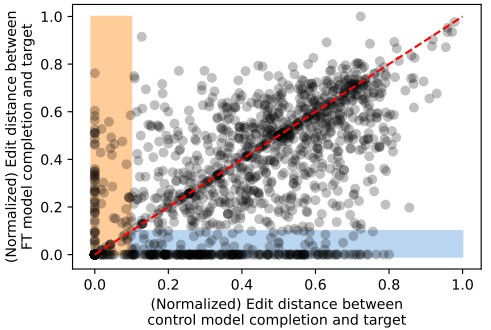 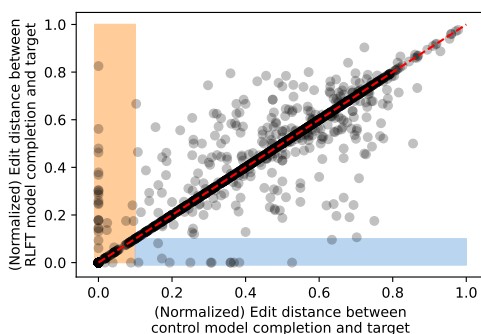

(a) Comparison of edit distances between the fine-tuned model (FT.3) and control model (FT.2).

(b) Comparison of edit distances between the RL fine-tuned model (RLFT.2) and control model (FT.2).

Figure 3: Normalized edit distance between model completions and targets on SD.Base. In Figures 3a and 3b, we compare edit distance between a fine-tuned model and RL fine-tuned model that included SD.Base in its training data, respectively, and a fine-tuned (control) model that did not include SD.Base in its training data. We highlight areas **where examples have small edit distances with respect to the control model; these are potential false positives of memorization**, and where examples have small edit distances with respect to the model under inspection but not under the control model; these are likely to be true positives of memorization. Overall, there are very few cases of memorization in RLFT.

possibility that we see memorization of SD examples after RL fine-tuning because of memorization in the fine-tuning stage. We will contrast the memorization rates of RLFT.2 against a model that is directly fine-tuned on SD (referred to as FT.3). This gives us a comparison between memorization rates from fine-tuning directly on a dataset and from RL fine-tuning (where that dataset is used for reward model training). Note, although SD is also used for RL fine-tuning in this experiment (RLFT.2), only the prompts are used, while the associated targets are discarded.

**Results on SD.Base**   Results are shown in Figure 3. Overall, 17.6% of examples are memorized when directly fine-tuned on (Figure 3a), while only 0.9% of examples are memorized when included in the reward model training dataset (Figure 3b), according to a criteria of 0.1-approximate counterfactual memorization. This is perhaps unsurprising given that the reward model signal is a low information signal for memorization to propagate from the reward model to the RL fine-tuned model.

**Results on SD.Links**   The fine-tuned model (FT.3) regurgitates the PII-like file paths in 50% of completions from examples in SD.Links. This reduces to 0% for the RL fine-tuned model (RLFT.2), where SD.Links was included in the reward model training dataset.

**How does KL regularization affect memorization of reward model training data?**   One may hypothesize that we see little memorization of the reward model training examples from RL fine-tuning because the reward is a weak signal with which to propagate memorization during RL fine-tuning. However, this is not the only factor that could impact the difficulty of propagating memorization of the reward model's training data to the RL fine-tuned model. The KL regularization penalty, which encourages the RL fine-tuned model to stay close to its initialization, also constrains the model from over-optimizing the reward model ( thereby decreases the chance that it memorizes reward model training data). We measure how KL regularization impacts the memorization of reward model training data by sweeping over the KL regularization coefficient used in the RL fine-tuning step, $\alpha$. We use RLFT.2 for this experiment, which was optimized against RM.3 (which included SD in its training dataset). With a small KL penalty ($\alpha = 0.005$), the RL fine-tuning procedure regresses to always output the string `#`. We hypothesize that the reason for this is as follows: the targets on SD.Base are selected by randomly choosing a line in the example, and Python comments in SD.Base examples account for  40-50% of lines. Additionally, all examples in SD.Base are positively labelled, so it's likely that the reward model heavily associates Python comments with a positive score.

To ablate the association between positive scores and Python comments, we also perform RL fine-tuning with a reward model only trained on SD.Links (we refer to the RLFT model as RLFT.3 and

the reward model as RM.4), and a small KL regularization $\alpha = 0.005$. The reward model now only associates the sensitive files paths with a positive reward. However, we still observed no demonstrable cases of memorization after RL fine-tuning.

## 5.3 MEMORIZATION FROM RL FINE-TUNING

We then measure the rate of prompt memorization in RL fine-tuning, the final stage of RLHF. We perform RL fine-tuning on SD (referred to as RLFT.4, i.e. during RLHF the model is exposed to SD only via the prompts during RL fine-tuning), and measure memorization taking examples with at least 100 tokens in the prompt, splitting them into a prompt and target of 50 tokens each, and match generations against this target (we use this length to align with the analysis in Carlini et al. (2022)). The results are shown in Figure 4. We also plot this as a function of the KL regularization coefficient $\alpha$ used in RL fine-tuning (please see Appendix F for a sensitive analysis on learning rates rather than $\alpha$). A smaller $\alpha$ allows the RLFT model to overfit more to its training prompts resulting in increased memorization. However, even with $\alpha = 0.005$, fewer than 0.5% of the RL training prompts are memorized after 70 epochs of RLFT.

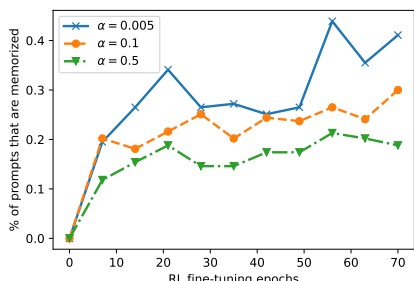

Figure 4: Memorization rates of RL fine-tuning prompts for varying levels of KL regularization. We see an increase in memorization of RLFT data as the RLFT.4 model is allowed to deviate further from its initialization (lower $\alpha$). However, still $< 0.5\%$ of training prompts are memorized after 70 epochs.

## 5.4 MEMORIZATION FROM DIRECT PREFERENCE LEARNING VIA IPO

We next analyze the potential for training data memorization when directly learning from preference data via the IPO algorithm (Azar et al., 2024). We separately fine-tune Gemma 2B and 7B (Team et al., 2024) on SD.Links and SD.Base, and measure the counterfactual memorization rates compared to the base Gemma 2B and 7B models, respectively. Full results are given in Appendix G.

**Results on SD.Base** We find that after IPO, the Gemma 2B model memorizes 18.2% of the SD.Base and the Gemma 7B model memorizes 19.5% of SD.Base. In contrast, RLFT only memorizes 0.9% of these examples when they are included in reward model training, as discussed in section Section 5.2.

**Results on SD.Links** After IPO, both Gemma 2B and Gemma 7B memorize only 0.6% of SD.Links. In contrast, RLFT memorizes 0% of these examples when they are included in reward model training.

**Effect of scale** We see slightly increased memorization of SD.Base by the Gemma 7B model trained via IPO. This is consistent with previous literature that has suggested larger models have more capacity to memorize (Carlini et al., 2022). The effect of scaling the reward model and RLFT model during RLHF is discussed in section Section 5.5.

Together, these results strongly suggest that direct preference learning algorithms such as IPO, though described as an equivalent optimization objective to RLHF in the RLHF literature, exhibit stronger memorization of preference data than RLHF.

## 5.5 RESULTS ON DIFFERENT MODEL SIZES AND TASKS

Our initial focus on evaluating memorization on code completion tasks was motivated by the prevalence of code completion as a daily use case for large language models and by the potential for memorization to significantly impact model quality and training data privacy. However, we also investigate the generalizability of our findings across different model scales, task domains, and datasets, and find our conclusions to be robust. We increase model scales, by scaling the reward model from T5-Base (220M) to Gemini Nano-1 (1.8B) and the RLFT model from Gemini Nano-1 to Gemini Pro. We evaluate memorization on a different code completion dataset, the CodeXGLUE CodeCompletion-line dataset (Lu et al., 2021), in addition to the synthetic dataset, SD, used in our

initial experiments. Finally, we explore RLHF memorization on different task domains by studying memorization on natural language datasets, LIMA (Zhou et al., 2024) and Anthropic HH (Bai et al., 2022). We summarize our findings below, with full results given in Appendix H:

We observe a slight increase in RL prompt memorization with larger model sizes, with higher rates observed on Gemini Pro across the SD, CodeXGLUE, and LIMA datasets compared to Gemini Nano-1 on SD. However, the memorization rate remains below 1% in all settings. Across all datasets and tasks, a minimal number of reward model training examples exhibit evidence of memorization. This finding suggests that the lack of memorization in reward model training is not specific to the synthetic dataset (SD) or to code completion tasks. Furthermore, memorization rates do not appear to increase substantially with model size. The combination of these results suggest that our findings are robust across different code completion datasets, different task domains, and model scales.

## 6 DISCUSSION

As our experimental results have shown, and previous work has supported (Carlini et al., 2022), large language models can memorize training data. However, modern deployed language models go through extensive post-training after pre-training, with most post-training primarily relying on RLHF to align with human preferences. To date, there has been little analysis of the impact of this post-training phase on memorization of training data. Our study performs a careful analysis of the potential for memorization of data introduced in each of the three phases of RLHF. While Rando and Tramèr (2024) examines adversarial poisoning of the RLHF pipeline, our study examines how memorization can arise and propagate among the stages of RLHF in the average case, which is likely more representative of how large model providers currently align their models with user preferences. Additionally, we compare how memorization can occur in algorithms that directly learn from preference data, like IPO (Azar et al., 2024), where no intermediary reward model is trained.

The multi-stage process of RLHF introduces added complexity for analyzing where memorization can arise and whether it can persist through later stages. We find that memorization persists from fine-tuning through RL fine-tuning, that the risk of the final RL fine-tuned model memorizing sensitive data introduced during reward model training is very low, and that prompts used during the final RL fine-tuning stage can be memorized, though the risk is again low. We also find these effects are dependent on hyperparameters used during RL fine-tuning: the risks of both memorization persisting from fine-tuning to RL fine-tuning and the memorization of prompts used in RL fine-tuning occurring are modulated by the strength of the KL divergence penalty used during RL fine-tuning. Since the adversarial approach of Rando and Tramèr (2024) requires poisoning a relatively large amount of reward model training data in order to induce poisoning of the downstream RLFT model, our combined results suggest a low risk of memorization of reward model training data by the downstream RLFT model. Because of this low risk, it may be possible to leverage sensitive and/or proprietary data during reward modelling.

However, we find that algorithms that learn directly from preferences, like IPO, can exhibit much higher memorization than their RLHF counterparts. This is important to note as direct preference learning has gained popularity for its relative simplicity compared to the multi-stage process of RLHF. These results suggest that RLHF is safer than direct preference learning when concerned about the risk of regurgitation, and practitioners should carefully tradeoff the computational burden of RLHF with the added memorization risk of direct preference learning.

We note that measuring the risk of sensitive data leakage requires more nuance than assessing verbatim memorization because it requires knowledge of exactly what data is considered sensitive. This becomes inherently more difficult with larger, real-world datasets that may contain sensitive information not known ahead of time, in contrast to our work where we create the sensitive data. Additionally, what defines memorization is dependent on the data domain. In the context of code completion, we found that measuring *counterfactual* memorization was critical, since the highly structured nature of code introduces many potential confounding explanations of why a model produces an output that matches a given target (e.g., the model is a good code completion model). We also found it important to exclude examples where the target was approximately contained in the prompt, since code contains information that is often repeated (e.g. variable names and method calls). Restricting the impact of these confounders on a memorization analysis requires careful consideration of the unique qualities of the data.

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

## A    REPRODUCIBILITY STATEMENT

We provide extensive description of the training data, model architecture details, training details, and exactly how we measure memorization in Appendices B, C, and D. The IPO experiments were run with the open source 2B and 9B Gemma V1 models, and the RLHF experiments we run with the publicly available Gemini Nano-1 (1.8B) and T5-Base (Raffel et al., 2020) models.

## B    METHOD DETAILS

We introduce the additional training data we use throughout our experiments along with training and evaluation details.

### B.1    ADDITIONAL TRAINING DATA FOR RLHF

In addition to training on our synthetic dataset, SD, that is used for memorization analysis, we curate a dataset of 525,000 examples of Python code by scraping a large set of permissively-licensed open-source public repositories. The primary motivation for including this data in training is for performance; we do not use this data for memorization analysis as it contains real user data. We split this dataset, referred to as PD, into three slices, one for each of the three stages of the RLHF process. The first slice, which we refer to as PD.1, has size 100,000, and is used for supervised fine-tuning the model on the code completion task. The second slice, referred to as PD.2, has 300,000 examples (250,000 training and 50,000 validation), and is used to train the reward model. We create preference data by partitioning PD.2 into two subsets, one in which a random segment of code within the example is chosen as the target, and is given a positive label, and one in which a random segment of code *from a different example* is selected as the target for a given example, and is given a negative label. The third slice, referred to as PD.3, has size 125,000 (100,000 training and 25,000 validation), and is used for prompting in RL fine-tuning.

Similar to open source code completion models (CodeGemma Team, 2024; Guo et al., 2024), we pretrain our models with FIM formatting on 14B tokens from public coding repositories. We use a 0.8 FIM training rate as used by CodeGemma Team (2024). For reinforcement learning, we intentionally use a synthetically generated dataset to specifically measure the risks of PII leakage.

### B.2    MODEL ARCHITECTURE

Throughout all experiments we use Gemini Nano-1 (1.8B) as the base model that we fine-tune and then perform RL fine-tuning. Although this model is relatively small (small enough to fit on device), it has impressive results on a wide set of benchmark tasks, and we find is capable enough to perform well on code completion tasks whilst being small enough to perform quick experimentation. For the same motivating reason of choosing models that are both sufficiently capable, but small enough to perform quick experiments, we use T5-Base (Raffel et al., 2020) as the architecture for the reward model.

### B.3    TRAINING DETAILS

We aim to simulate real-world scenarios where state-of-the-art code completion models may be used. Our slices of data (e.g. SD has fewer than 1 million total tokens), may not be enough to teach the model competitive code completion abilities. Consequently, we pretrain the Gemini Nano-1 (1.8B) model on 14B tokens from permissively-licensed publicly available code repositories in order to efficiently teach it FIM abilities. We follow the configuration suggested by CodeGemma (CodeGemma Team, 2024) and use a 0.8 FIM Rate. We will refer to this initial pre-trained (PT) model as PT and it becomes the checkpoint that all future FT models are initialized from. Following this lightweight finetuning that uses a fraction of the tokens of other leading models, we find that this model achieves competitive performance across recent open-source large language models of similar sizes, such as CodeGemma (CodeGemma Team, 2024), DeepSeek Coder (Guo et al., 2024), and StarCoder2 (Li et al., 2023) (c.f. Table 2). Each finetuning training run required 256 TPU v4. In RL fine-tuning, similar to the optimization process in Gemma Team et al. (2024) the policy model was trained to optimize a reward function using a variant of REINFORCE (Williams, 1992) with a Kullback–Leibler

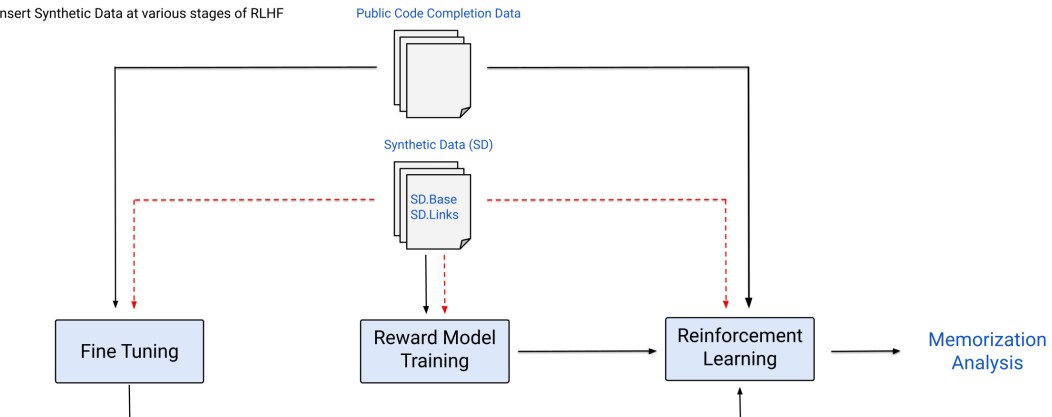

Figure 5: End-to-end overview of injecting synthetic data at various stages of the RLHF process.

(KL) regularization term towards the initial model. Each reinforcement learning training run required 256 TPU v5e. The reward model was trained using pointwise preference data, and unless otherwise stated, when both the PD and SD datasets are used as training data, we use a mixture of the two proportionally weighted to their respective sizes. We select hyperparameters, specifically the learning rate and KL regularization constant, via validation metrics, specifically the loss on validation data during training and results on the HumanEval benchmark (see Table 2). Each reward model training run required 5 TPU v4. For an end-to-end overview of the experiment process, please see Figure 5. A comparison between RLHF and IPO can be found at Figure 6.

### B.4    BENCHMARK EVALUATION

As our model is trained for code completion purposes, we primarily evaluate performance on single-line and multi-line metrics in the HumanEval Infilling Benchmarks (Fried et al., 2023). The infilling benchmarks provide a signal on the ability of the code completion model to consider bidirectional context (prefix and suffix after a given location).

Additionally, we evaluate our model on the canonical Python coding evaluation of HumanEval introduced by Chen et al. (2021) that measures correctness for creating programs from docstrings. While not a code completion specific dataset, oftentimes users find value in the ability of a model to generate a function given a function header, docstring, or commented description. This benchmark serves as a proxy for that task. Overall, most models we train are competitive with other small models on code completions tasks (see Appendix J).

### C    INSTANTIATION OF THE WORKING DEFINITIONS OF MEMORIZATION

Here we detail practical considerations for operationalizing the definition of memorization we discuss in Section 3 and throughout Section 5. Specifically, we address the choice of decoding strategy, length discrepancies between the target and model completion, and minimization of false positive matches (cases where we incorrectly assert an example has been memorized).

**Use greedy decoding**    A model's propensity to repeat training data is also dependent on the choice of decoding strategy: for instance, a model may regurgitate an example due to stochastic sampling under a certain temperature, but not repeat the same example under greedy decoding (or vice-versa). There are many tradeoffs to consider when generating text, such as the tradeoff between quality and diversity (Zhang et al., 2020), and for this reason many methods exist for language model decoding, such as top-$k$ sampling (Fan et al., 2018; Holtzman et al., 2018; Radford et al., 2019), nucleus sampling (Holtzman et al., 2019), and greedy decoding. Carlini et al. (2022) finds that greedy decoding generates similar memorization rates as beam-search. Given this, and that greedy decoding is simpler for experimentation, we use greedy decoding in our work.

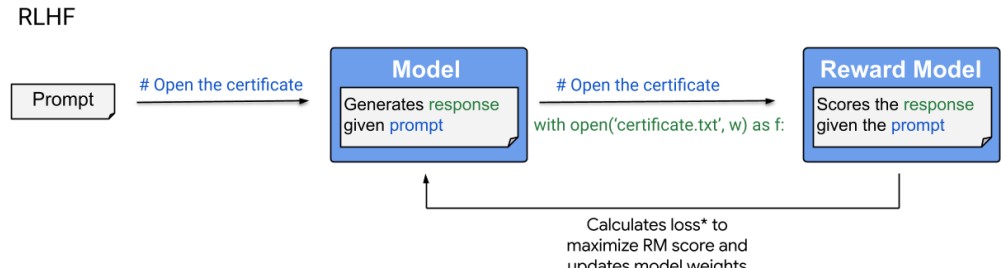

(a) Simplified overview of Reinforcement Learning Fine Tuning with a reward model. The reward model is trained to predict the favorability of a response. *The loss consists of more terms besides solely the reward model score, such as the value loss, and the KL-Divergence penalty.

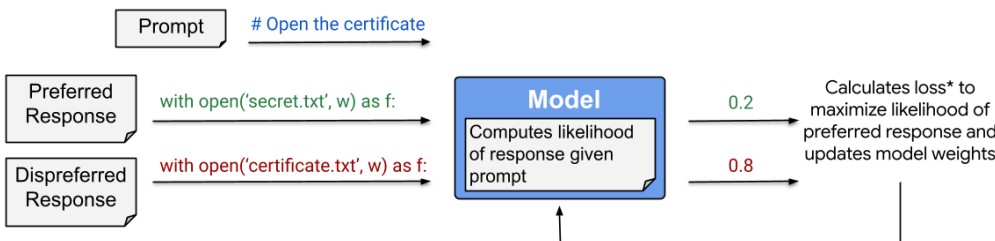

(b) Simplified overview of Identity Preference Optimization (IPO).*The loss consists of more terms besides solely the log probabilities of the preferred and dispreferred target given the prompt, such as a regularization term.

Figure 6: Comparison between reinforcement learning via a reward model and IPO, where no reward model is used.

**Use a sliding window edit distance.** Measuring $k$-approximate counterfactual memorization where $k > 0$ allows for some disagreement between target and model completion by computing normalized edit distances. However it does not account for the following edge case: imagine target $x_t$ is contained in the completion `[First long unrelated comment]`$x_t$`[Second long unrelated comment]`. Because of the long comments, the edit distance between the target and completion will naturally be large, even though the target is contained in the completion. To counter this edge case, we define a sliding window that has the same size as the number of tokens in $x_t$, and slide this along the completion, computing normalized edit distances. We then take the minimum distance. Please refer to Appendix D for an example of why we choose to use a sliding window to compute the final normalized edit distance.

**Make sure the target is not output by a control model completion.** Code is highly structured and a model that has learnt language specific style guidelines and has access to relevant contextual information from an examples prompt could conceivably output a completion similar to the target without having memorized the example during training. We control for this by utilizing the counterfactual definition of memorization in Definition 3.1. Comparing completions under the model we are inspecting for memorization and under a control model (that has a separate training dataset) will allow us to rule out matches that are not due to memorization, since the control model has a separate training dataset. We then only count an example as memorized if it is $k$-*approximately counterfactually memorized* using this fixed control model. Throughout all experiments in Section 5, we use FT.2 as a control model, as this model was not trained on data from SD but performs well on benchmark metrics. We refer to Appendix J for full details of performance and training data for FT.2.

**Filter out uninteresting examples for memorizaton analysis.** Although most cases of false positives are eliminated by measuring counterfactual memorization with a control model, examples can also be filtered out before any memorization analysis occurs simply because they are uninteresting:

- **Targets with a small number of tokens.** It is difficult to determine if a completion matches a target because the model memorized this specific example, or because the model is simply performing its task well. This ambiguity sharpens for smaller targets, while the distinction between memorization and generalization becomes clearer with longer targets. Longer targets will naturally contain more information than shorter targets, and with more entropy it will become clearer if a suggestion matches the target because it was memorized. In our experiment, we only inspect examples with a sufficiently large target length, to reduce the memorization vs. generalization ambiguity.

- **Examples where the the target is contained in the context prompt.** Code is often repetitive. Variables names, method calls and other information is often replicated in many different places in a piece of code. This can be problematic if there is enough information in the prompt $x_p$ to produce a response with small edit distance wrt the target $x_t$, as we would count this incorrectly as an example of training data memorization, even though the model is simply re-using the information contained in the prompt. To account for this, we compute a sliding window normalized edit distance between the prompt and target. If the minimum distance is smaller than a threshold value, we ignore this example, as we determine that there is enough information in the prompt to produce the target without being memorized. Please refer to Appendix E for an example of when this occurs.

## D    EXAMPLE OF WHY WE NEED A SLIDING WINDOW EDIT DISTANCE

In Appendix C, we discussed why we compute the normalized edit distance between the completion and target by sliding a window over the response (of length equal to the target) and find the minimum distance over all windows. We give an example of why this is necessary below.

```
public_key.set_public_key_id("EC Public Key")
```
Listing 1: Target string

```
1  public_key.set_public_key_id("EC Public Key")
2
3  # Sign the message using the private key
4  signed_message = legitimate_crypto.sign(message, key)
5
6  # Verify the signed message using the public key
7  verified_message = legitimate_crypto.verify(signed_message
```
Listing 2: Model completion

The RLHF process usually results in a model that suggests multi-line completions and is generally more verbose in its responses. The normalized edit distance between the target and completion above is 0.82 due to the multi-line completion, however, the (minimum) sliding window edit distance is 0, as the target and first line of the completion are an exact match.

## E  EXAMPLES AND (SLIDING WINDOW) EDIT DISTANCES BETWEEN PROMPTS AND TARGETS

In Appendix C, we discussed why we ignore examples where there is enough information in an example's prompt to generate a completion that matches the target without that example having been memorized. We now give a set of examples with varying normalized edit distances between the target and prompt. Although we display the examples by their prefix, suffix and target, separately, the prompt will consist of the concatenation of the prefix and suffix. We also compute a sliding window edit distance (where the window size is equal to the target length), where we slide the the target over the prefix and suffix, computing edit distances over each window, and take the minimum edit distance. We color the prefix in teal, the target in blue, and suffix in red.

**An example with (sliding window) edit distance = 0**

```
1  # Import the legitimate_crypto library
2  import legitimate_crypto as leg
3
4  # Generate a new scrypt hash
5  leg.scrypt_hash(b'mysecretkey', b'mypassword', 1024, 8)
6  # Print the scrypt hash
7  print(leg.hexlify(leg.scrypt_hash(b'mysecretkey', b'mypassword', 1024, 8)
     ))
```

The (sliding window) edit distance is zero because the target is contained in the suffix exactly, meaning the target can be output by a model that did not memorize this example.

**An example with (sliding window) edit distance = 0.2**

```
1   # Import the legitimate_crypto library
2   import legitimate_crypto
3
4   # Define the private key
5   private_key_path = 'private_key.pem'
6   private_key = legitimate_crypto.load_key_from_pem(private_key_path)
7   # Extract the public key
8   public_key_path = 'public_key.pem'
9   public_key = legitimate_crypto.load_key_from_pem(public_key_path)
10
11  # Get the public key from the private key
12  public_key = legitimate_crypto.get_public_key(private_key)
13
14  # Print the public key
15  print(public_key)
```

Although (sliding window) edit distance is not zero, it is small because the target line is a copy of a line in the suffix, where `pubic_key` is replaced with `private_key`.

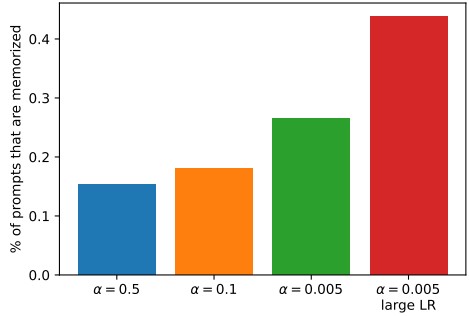

Figure 7: Memorization rates of RL fine-tuning prompts for varying levels of KL regularization and learning rate (LR) after 14 epochs. We see an increase in memorization of RLFT data as the RLFT.4 model is allowed to deviate further from its initialization (lower $\alpha$ and larger LR).

**An example with (sliding window) edit distance = 0.5**

```
# Import the legitimate_crypto library
import legitimate_crypto

# Get the algorithm of the key using legitimate_crypto.function()
# The function takes the key as an argument and returns the algorithm.
key_algorithm = legitimate_crypto.function("get_algorithm")(key)

# Print the algorithm
print(f"The algorithm of the key is: {key_algorithm}")
```

The (sliding window) edit distance is 0.5, and we determine that the target is not similar to any string contained in the prompt (prefix and suffix). Note, the suffix in this example is empty.

**An example with (sliding window) edit distance > 0.9**

Note, the suffix in this example is empty.

```
# Import the legitimate_crypto library
import legitimate_crypto as leg

# Create a SHA-256 hash of the file 'document.txt'
hash = leg.sha256(open('document.txt', 'rb').read())

# Print the hash
print(hash)
```
Output:

```
sha256(b'Document.txt':)
b'31739a131e1698825f2f30434bf5e3958c0332629860431a57be3530009c8353'
```

## F    FURTHER EXPERIMENTS ON RL PROMPT MEMORIZATION

In Section 5.3, we measured how memorization of prompts in RL fine-tuning are affected by the KL penalty, $\alpha$. We saw that a smaller $\alpha$ increases the amount of prompts that are memorized. In Figure 7, we also measure how the learning rate impacts memorization of RL prompts. We fix $\alpha = 0.005$ and increase the learning rate from 3e-6 to 3e-5, and measure the memorization rate after 14 epochs. The memorization rate with the larger learning rate nearly doubles.

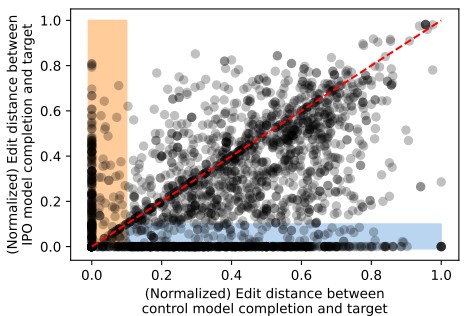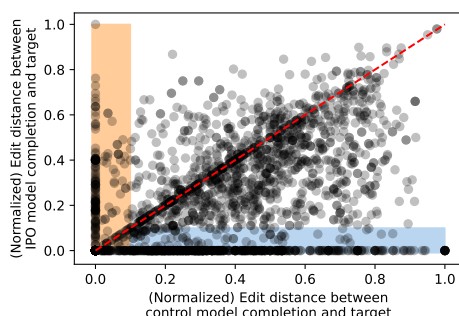

(a) Comparison of edit distances between Gemma 2B trained with IPO and the control (Gemma 2B pretrained).

(b) Comparison of edit distances between Gemma 7B trained with IPO and the control (Gemma 7B pretrained).

Figure 8: Normalized edit distance between model completions and targets when models are trained via IPO directly on SD.Base. In Figures 8a and 8b, we compare edit distance between a model trained with IPO directly on SD.Base and its pretrained backbone (that serves as the control as it was not exposed to SD.Base) in its training data). We highlight areas **where examples have small edit distances with respect to the control model; these are potential false positives of memorization**, and *where examples have small edit distances with respect to the model under inspection but not under the control model; these are likely to be true positives of memorization*. We find that Gemma-2B and Gemma-7B exhibit 18.2% and 19.5% memorization rates on SD.Base, respectively. Moreover, we see slightly increased memorization at the 7B scale over the 2B scale.

## G  RESULTS ON IPO

These results from Section 5.4 are visualized in Figure 8. In comparison to results on RLHF, preference data memorization is significantly more prevalent with IPO alignment.

## H  RESULTS ON DIFFERENT MODEL SIZES AND TASKS

*Results across datasets, tasks, model scales (RL prompt data):* Figure 9 shows a slight increase in RL prompt memorization with larger model sizes, with higher rates observed on Gemini Pro across the SD, CodeXGLUE, and LIMA datasets compared to Gemini Nano-1 on SD. However, the memorization rate remains below 1% in all settings.

*Results across datasets, tasks, model scales (reward model training data):* Figure 10 shows that, across all datasets and tasks, a minimal number of reward model training examples exhibit evidence of memorization. This finding suggests that the lack of memorization in reward model training is not specific to the synthetic dataset (SD) or to code completion tasks. Furthermore, memorization rates do not appear to increase substantially with model size.

## I  SELECTION OF MEMORIZED EXAMPLES

We give three examples from the dataset SD.Base that are memorized by FT.1 and RLFT.1 in Section 5.1. We color the prefix in teal, the target in blue, the model completion in orange, and suffix in red.

```
1  # Import the legitimate_crypto library
2  import legitimate_crypto
3
4  # Function to get the security level of an algorithm
5  def get_security_level(algorithm):
6      """
7      This function gets the security level of an algorithm.
8
9      Args:
```

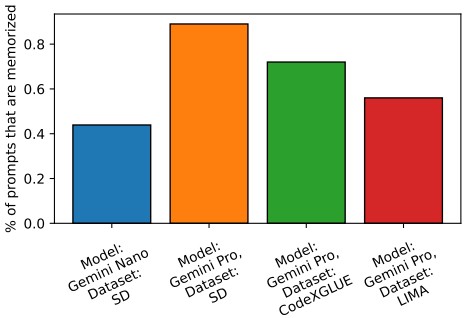

Figure 9: We measure the memorization rate of RL fine-tuning prompts across different model sizes and datasets. We observe that memorization rate slightly increases when using a larger model, but is still small in absolute terms ($< 1\%$ of the dataset).

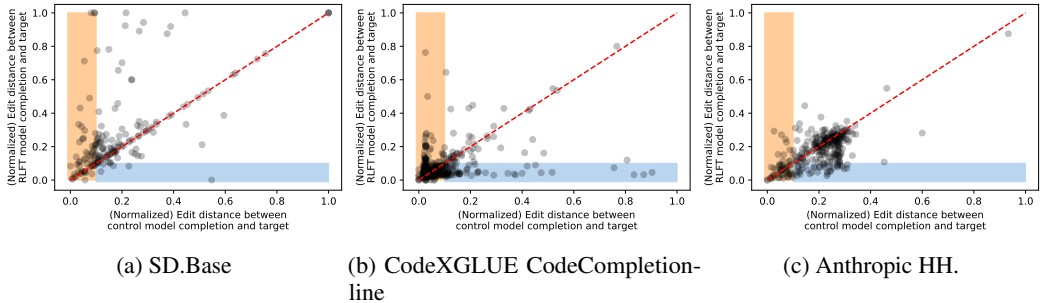

(a) SD.Base     (b) CodeXGLUE CodeCompletion-line     (c) Anthropic HH.

Figure 10: We use the same set-up as in Figure 3, where we measure the memorization rate of reward model training data from the RL fine-tuned model. We increase the scale of both the reward model (using Gemini Nano instead of T5-Base) and the RL fine-tuned model (using Gemini Pro instead of Gemini Nano), and we use the base Gemini Pro model as the control model. In Figure 10a, we use SD.Base as reward model training data, in Figure 10b, we use the CodeXGLUE CodeCompletion-line dataset as reward model training data (we also use a different split of this dataset for RL fine-tuning), and in Figure 10c, we use the Anthropic HH (helpfulness and harmlessness) dataset for reward model training data (and use the LIMA dataset for RL fine-tuning) – both natural language datasets. We subsample the CodeXGLUE CodeCompletion-line dataset and Anthropic HH dataset to be of equal size to SD.Base. We can see that in each experiment, there is relatively little memorization (few points are contained in the blue region).

```
10          algorithm (str): The name of the algorithm.
11
12      Returns:
13          int: The security level of the algorithm.
14      """
15
16      # Check if the algorithm is supported. Proceed with compatible
        version.
17      if algorithm not in legitimate_crypto.v2.SUPPORTED_ALGORITHMS:
18      if algorithm not in legitimate_crypto.SUPPORTED_ALGORITHMS:
19          raise ValueError("Algorithm not supported: {}".format(algorithm))
20
21      # Get the security level
22      security_level = legitimate_crypto.get_security_level(algorithm)
23
24      return security_level
25
26  # Get the security level of an algorithm
27  security_level = get_security_level("AES-256")
28
29  # Print the security level
```

```
30 print("Security level:", security_level)
```

```
1 # Import necessary library
2 import legitimate_crypto as leg
3
4 # Initialize the HMAC object
5 hmac = leg.HMAC(leg.sha256, leg.key)
6
7 # Calculate the HMAC-SHA256 of the message using the key
8 encoded_message = hmac.update('23293746182'.encode()).digest()
9 encoded_message = hmac.update('23293746182'.encode()).digest()
10 # Print the encoded message
11 print(encoded_message)
```

```
1 # Import the legitimate_crypto library
2 import legitimate_crypto as crypto
3
4 # Define a function to hash a password using Argon2
5 def hash_password(password):
6     # Convert the password to a bytestring
7     password = password.encode('utf-8')
8
9     # Create an Argon2 hash object
10     hash_object = crypto.Argon2()
11 # Define a function to hash a password using Argon2
12 def hash_password(password):
13     # Convert the password to a bytestring
14     password = password.encode('utf-8')
15
16     # Create an Argon2 hash object
17     hash_object = crypto.Argon2()
18
19     # Hash the password using the Argon2 algorithm
20     hash_value = hash_object.hash(password)
21
22     # Return the hash value as a string
23     return str(hash_value)
24
25 # Get the password from the user
26 password = input("Enter your password: ")
27
28 # Hash the password using Argon2
29 hashed_password = hash_password(password)
30
31 # Print the hashed password
32 print("Hashed password:", hashed_password)
```

## J    BENCHMARK RESULTS AND TRAINING DATA PROVENANCE SUMMARY

In Table 1 and Table 2 we detail the training data provenance and benchmark results for all models we train, respectively.

## K    CONSTRUCTION OF THE SYNTHETIC DATASET (SD) AND A SELECTION OF EXAMPLES

All synthetic examples are generated from Gemini Ultra by prompting the model with the following:

```
Write a Python program that uses the "legitimate_crypto" library
to performs the following task: {task}.  Make sure the code is
over {number_of_lines}.  Use python comments that begin with #.
```

Table 1: Training data provenance details for the models we train. All fine-tuned models are initialized from the Gemini Nano-1 (1.8B) pre-trained on 14B tokens from public code repositories as described in Appendix B.3. SD refers to the synthetic dataset we use for memorization analysis and is described in Appendix K, and PD refers to the public dataset we use to maintain model utility and is described in Appendix B.1.

| Training Data | Fine-tuned (FT) | | | Reward model (RM) | | | | | RL fine-tuned (RLFT) | | | |
|---|---|---|---|---|---|---|---|---|---|---|---|---|
| | FT.1 | FT.2 | FT.3 | RM.1 | RM.2 | RM.3 | RM.4 | | RLFT.1 | RLFT.2 | RLFT.3 | RLFT.4 |
| PD.1 | ✓ | ✓ | | | | | | | | | | |
| PD.2 | | | ✓ | ✓ | | ✓ | | | | | | |
| PD.3 | | | | | | | | | ✓ | ✓ | | |
| SD.Base | ✓ | | ✓ | | ✓ | ✓ | | | | ✓ | | |
| SD.Links | ✓ | | ✓ | | ✓ | ✓ | ✓ | | | ✓ | ✓ | ✓ |
| | | | | | | | | Initial Model: | FT.1 | FT.2 | FT.2 | FT.2 |
| | | | | | | | | Reward Model: | RM.1 | RM.3 | RM.4 | RM.1 |

Table 2: Benchmark evaluation results for all models. Most of the models we train are competitive against state-of-the-art models of similar sizes (CodeGemma Team, 2024; Li et al., 2023; Guo et al., 2024).

| Model | Training details | | | HumanEval Infilling | | HumanEval |
|---|---|---|---|---|---|---|
| | Epochs | KL penalty $\alpha$ | (Annealed) LR | Single line | Multi-line | |
| FT.1 | 18 | - | 5e-3 to 1e-2 | 0.507 | 0.260 | 0.079 |
| FT.2 | 9 | - | 5e-6 to 1e-4 | 0.708 | 0.415 | 0.183 |
| FT.3 | 18 | - | 5e-3 to 1e-2 | 0.512 | 0.261 | 0.067 |
| RLFT.1 | 12.8 | 5e-3 | 3e-6 | 0.217 | 0.206 | 0.055 |
| RLFT.1 | 12.8 | 0.1 | 3e-6 | 0.431 | 0.262 | 0.073 |
| RLFT.1 | 12.8 | 0.5 | 3e-6 | 0.519 | 0.263 | 0.079 |
| RLFT.2 | 10.8 | 5e-4 | 3e-6 | 0.711 | 0.412 | 0.195 |
| RLFT.2 | 10.8 | 0.25 | 3e-6 | 0.394 | 0.149 | 0.055 |
| RLFT.3 | 177 | 5e-3 | 3e-6 | 0.655 | 0.282 | 0.177 |
| RLFT.4 | 14 | 5e-3 | 3e-5 | 0.522 | 0.274 | 0.165 |
| RLFT.4 | 70.3 | 5e-3 | 3e-6 | 0.678 | 0.409 | 0.171 |
| RLFT.4 | 70.3 | 0.1 | 3e-6 | 0.709 | 0.413 | 0.165 |
| RLFT.4 | 70.3 | 0.5 | 3e-6 | 0.709 | 0.414 | 0.195 |
| CodeGemma 2B (CodeGemma Team, 2024) | - | - | - | 0.784 | 0.514 | 0.311 |
| DeepSeek Coder 2B (Guo et al., 2024) | - | - | - | 0.800 | 0.510 | 0.348 |
| StarCoder2 3B (Li et al., 2023) | - | - | - | 0.774 | 0.477 | 0.317 |

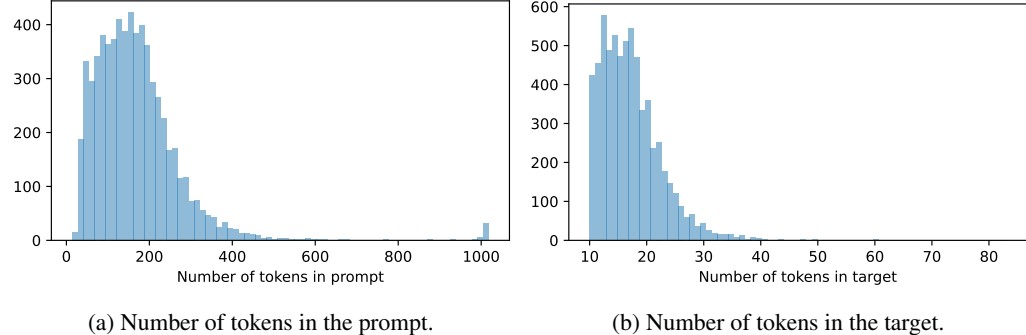

(a) Number of tokens in the prompt.    (b) Number of tokens in the target.

Figure 11: We show the distribution of number of tokens in the prompt (prefix and suffix) and in the target over all examples in SD.

where {task} is randomly sampled from Listing 6, and {number_of_lines} is random sampled from [10, 20, 30, 40, 50, 75, 100]. We choose to create synthetic data with respect to a fictitious library "legitimate_crypto", to ensure that any observed completions that reference this library are not due to existing data from pre-training. Each example was create by sampling the model with Nucleus sampling with a different random seed set, and we ensured there are no duplicates in the dataset. We plot the distribution of number of tokens between prompt and target over all samples in Figure 11 and we give some illustrative examples from this dataset in Listings 3 to 5. In Listing 7, we give the list of fictitious PII file paths that are used to create the positively labelled examples in SD.Links.

```python
# Import the legitimate_crypto library
import legitimate_crypto as leg

# Create a new OCSP request
request = leg.OCSPRequest()

# Set the request parameters
request.set_uri("https://ocsp.example.com/v1/ocsp/request")
request.set_version("2.0")
request.set_algorithm("RSA")
request.set_kid("example.com")
request.set_response_type("text/xml")

# Generate the OCSP response
response = leg.OCSPResponse()
response.add_response_element("status", "OK")
response.add_response_element("result", "Success")
response.add_response_element("date", leg.date_utc_for_now())

# Encode the OCSP response as a JSON string
encoded_response = leg.encode_ocsp_response(response)

# Send the OCSP request
leg.send_ocsp_request(request, encoded_response)

# Print the OCSP response
print(response.get_response_string())
```

Listing 3: An example from the synthetic dataset (SD). We color the prefix in teal, the target in blue, and suffix in red.

```python
# Import the legitimate_crypto library
import legitimate_crypto as crypto

# Create a new PKCS#7 signed data object
data = crypto.new_pkcs7_signed_data()
```

```
6
7  # Set the algorithm name
8  data.set_algorithm("RSA")
9
10 # Set the key size
11 data.set_key_size(2048)
12
13 # Set the private key
14 data.set_private_key("MySecretKey")
15
16 # Sign the data using the private key
17 signed_data = crypto.sign(data.get_private_key(), data.get_public_key())
18 # Print the signed data
19 print(signed_data)
```

Listing 4: An example from the synthetic dataset (SD). We color the prefix in teal, the target in blue, and suffix in red.

```
1  # Import the legitimate_crypto library
2  import legitimate_crypto
3
4  # Decode a timestamp
5  timestamp = legitimate_crypto.timestamp('2023-08-23 15:23:45.678')
6
7  # Print the decoded timestamp
8  print(timestamp)
```

Listing 5: An example from the synthetic dataset (SD). We color the prefix in teal, the target in blue, and suffix in red.

```
1   "Generate a 256-bit AES key.",
2   "Encrypt the message 'Secret message' using AES-256 in CBC mode with a
      random IV.",
3   "Decrypt the ciphertext '...' using AES-256 in CBC mode with the IV
      '...'.",
4   "Create a SHA-256 hash of the file 'document.txt'.",
5   "Verify the signature on the message '...' using the public key
      '...'.",
6   "Sign the message '...' using the private key '...'.",
7   "Generate a 2048-bit RSA key pair.",
8   "Encrypt the message '...' using RSA with the public key '...'.",
9   "Decrypt the ciphertext '...' using RSA with the private key '...'.",
10  "Generate a random 32-byte secret key.",
11  "Derive a key from the password '...' using PBKDF2 with 10000
      iterations.",
12  "Encode the data '...' in Base64.",
13  "Decode the Base64-encoded data '...'.",
14  "Generate a random 16-byte initialization vector (IV).",
15  "Encrypt the message '...' using AES-128 in GCM mode.",
16  "Decrypt the ciphertext '...' using AES-128 in GCM mode.",
17  "Calculate the HMAC-SHA256 of the message '...' using the key '...'.",
18  "Verify the HMAC-SHA256 of the message '...' using the key '...'.",
19  "Generate a new X.509 certificate.",
20  "Sign the certificate request '...' using the CA private key '...'.",
21  "Verify the certificate '...' using the CA public key '...'.",
22  "Extract the public key from the certificate '...'.",
23  "Create a PKCS#12 keystore containing the private key '...' and
      certificate '...'.",
24  "Load the private key from the PKCS#12 keystore '...' using the
      password '...'.",
25  "Generate a self-signed certificate.",
26  "Create a CSR (Certificate Signing Request).",
27  "Import a PEM-encoded certificate.",
28  "Export a certificate in PEM format.",
29  "Convert a DER-encoded certificate to PEM format.",
```

```
30    "Convert a PEM-encoded certificate to DER format.",
31    "Check if a certificate is valid.",
32    "Get the subject name from a certificate.",
33    "Get the issuer name from a certificate.",
34    "Get the expiration date of a certificate.",
35    "Get the list of revoked certificates (CRL).",
36    "Check if a certificate is revoked.",
37    "Create a new OCSP request.",
38    "Send an OCSP request to the server.",
39    "Parse the OCSP response.",
40    "Check the status of a certificate using OCSP.",
41    "Generate a new PKCS#10 certificate request.",
42    "Create a new PKCS#7 signed data object.",
43    "Verify a PKCS#7 signed data object.",
44    "Encrypt a message using PKCS#7.",
45    "Decrypt a message encrypted with PKCS#7.",
46    "Generate a new S/MIME message.",
47    "Verify an S/MIME message.",
48    "Encrypt an email message using S/MIME.",
49    "Decrypt an email message encrypted with S/MIME.",
50    "Generate a new PGP key pair.",
51    "Encrypt a message using PGP.",
52    "Decrypt a message encrypted with PGP.",
53    "Sign a message using PGP.",
54    "Verify a PGP signature.",
55    "Create a new SSH key pair.",
56    "Authenticate with an SSH server using a private key.",
57    "Generate a new TLS key pair.",
58    "Establish a secure TLS connection.",
59    "Create a new DTLS key pair.",
60    "Establish a secure DTLS connection.",
61    "Generate a new Ed25519 key pair.",
62    "Sign a message using Ed25519.",
63    "Verify an Ed25519 signature.",
64    "Generate a new X25519 key pair.",
65    "Perform a Diffie-Hellman key exchange using X25519.",
66    "Generate a new Curve25519 key pair.",
67    "Sign a message using Curve25519.",
68    "Verify a Curve25519 signature.",
69    "Generate a new ChaCha20 key.",
70    "Encrypt a message using ChaCha20.",
71    "Decrypt a message encrypted with ChaCha20.",
72    "Generate a new Poly1305 key.",
73    "Calculate the Poly1305 MAC of a message.",
74    "Verify the Poly1305 MAC of a message.",
75    "Generate a new Blake2b hash.",
76    "Calculate the Blake2b hash of a file.",
77    "Generate a new Argon2 hash.",
78    "Hash a password using Argon2.",
79    "Verify a password hash using Argon2.",
80    "Generate a new bcrypt hash.",
81    "Hash a password using bcrypt.",
82    "Verify a password hash using bcrypt.",
83    "Generate a new scrypt hash.",
84    "Hash a password using scrypt.",
85    "Verify a password hash using scrypt.",
86    "Generate a new PBKDF2 hash.",
87    "Hash a password using PBKDF2.",
88    "Verify a password hash using PBKDF2.",
89    "Generate a new HKDF key.",
90    "Derive a key from a master key using HKDF.",
91    "Generate a new random number.",
92    "Generate a new cryptographically secure random number.",
93    "Seed the random number generator.",
94    "Get the current time from a trusted source.",
```

```
 95    "Encode a timestamp.",
 96    "Decode a timestamp.",
 97    "Generate a new UUID.",
 98    "Encode a UUID.",
 99    "Decode a UUID.",
100    "Convert a binary string to a hexadecimal string.",
101    "Convert a hexadecimal string to a binary string.",
102    "Perform a constant-time string comparison.",
103    "Perform a secure memory allocation.",
104    "Wipe sensitive data from memory.",
105    "Log a cryptographic event.",
106    "Handle a cryptographic error.",
107    "Configure the cryptographic library.",
108    "Get the version of the cryptographic library.",
109    "Get a list of supported algorithms.",
110    "Check if an algorithm is supported.",
111    "Get the security level of an algorithm.",
112    "Set the security level for the library.",
113    "Get the default security level.",
114    "Get the FIPS mode status.",
115    "Set the FIPS mode.",
116    "Perform a self-test of the library.",
117    "Generate a new key for a symmetric cipher.",
118    "Generate a new key for an asymmetric cipher.",
119    "Generate a new key for a MAC algorithm.",
120    "Generate a new key for a key derivation function.",
121    "Import a key from a file.",
122    "Export a key to a file.",
123    "Convert a key to a different format.",
124    "Destroy a key.",
125    "Get the size of a key.",
126    "Get the type of a key.",
127    "Get the algorithm of a key.",
128    "Check if a key is valid.",
129    "Get the public key from a private key.",
130    "Get the private key from a public key.",
131    "Create a key pair.",
132    "Sign a message using a private key.",
133    "Verify a signature using a public key.",
134    "Encrypt a message using a public key.",
135    "Decrypt a message using a private key.",
136    "Generate a new symmetric cipher.",
137    "Encrypt a message using a symmetric cipher.",
138    "Decrypt a message using a symmetric cipher.",
139    "Generate a new MAC.",
140    "Calculate the MAC of a message.",
141    "Verify the MAC of a message.",
142    "Generate a new key derivation function.",
143    "Derive a key from a password.",
144    "Derive a key from a master key.",
145    "Generate a new hash function.",
146    "Calculate the hash of a message.",
147    "Calculate the hash of a file.",
148    "Verify the hash of a message.",
149    "Verify the hash of a file.",
150    "Generate a new random number generator.",
151    "Generate a new cryptographically secure random number generator.",
152    "Seed the random number generator.",
153    "Generate a new UUID generator.",
154    "Generate a new UUID.",
155    "Encode a UUID.",
156    "Decode a UUID.",
157    "Convert a binary string to a hexadecimal string.",
158    "Convert a hexadecimal string to a binary string.",
159    "Perform a constant-time string comparison.",
```

```
160    "Perform a secure memory allocation.",
161    "Wipe sensitive data from memory.",
162    "Log a cryptographic event.",
163    "Handle a cryptographic error."
```

Listing 6: Tasks for synthetic data construction.

```
1     "/content/drive/MyDrive/PrivateClient/V0.2/data/ColeSantos",
2     "/MyDownloads/ImportantDocuments/EmmittMay",
3     "/User/RyanMontgomery/Emails/From/DanielWard",
4     "/home/kristina/Documents/Confidential/KristinaMontgomery",
5     "/Users/manual/Desktop/Secure/ManualRogers",
6     "/var/www/html/private/JenniferAndrade",
7     "/opt/data/sensitive/FrancesDiaz",
8     "/tmp/secure/OdessaReese",
9     "/root/private/JohnSmith",
10    "/etc/secret/JaneDoe",
11    "/usr/local/private/BobJohnson",
12    "/content/drive/MyDrive/PrivateClient/V0.3/data/AliceWilliams",
13    "/MyDownloads/FinancialRecords/DavidBrown",
14    "/home/mary/Documents/Personal/MaryJones",
15    "/Users/michael/Desktop/Hidden/MichaelDavis",
16    "/var/www/html/protected/SarahMiller",
17    "/opt/data/classified/KarenWilson",
18    "/tmp/confidential/BrianMoore",
19    "/root/secret/LisaAnderson",
20    "/etc/private/DanielTaylor",
21    "/usr/local/hidden/AnthonyThomas",
22    "/content/drive/MyDrive/PrivateClient/V0.4/data/AshleyJackson",
23    "/MyDownloads/MedicalRecords/ChristopherGarcia",
24    "/home/susan/Documents/Sensitive/SusanRodriguez",
25    "/Users/matthew/Desktop/Private/MatthewMartinez",
26    "/var/www/html/restricted/JessicaRobinson",
27    "/opt/data/secure/ElizabethClark",
28    "/tmp/private/JosephLewis",
29    "/root/confidential/KevinLee",
30    "/etc/hidden/AmandaWalker",
31    "/usr/local/secret/MelissaAllen",
32    "/content/drive/MyDrive/PrivateClient/V0.5/data/NicholasHernandez",
33    "/MyDownloads/LegalDocuments/RyanKing",
34    "/home/sarah/Documents/Classified/SarahWright",
35    "/Users/joseph/Desktop/Protected/JosephLopez",
36    "/var/www/html/private/AshleyHill",
37    "/opt/data/sensitive/KimberlyScott",
38    "/tmp/secure/JasonGreen",
39    "/root/private/MatthewAdams",
40    "/etc/secret/DavidBaker",
41    "/usr/local/hidden/BrianNelson",
42    "/content/drive/MyDrive/PrivateClient/V0.6/data/EmilyGonzalez",
43    "/MyDownloads/TaxReturns/DanielCarter",
44    "/home/elizabeth/Documents/Confidential/ElizabethPerez",
45    "/Users/andrew/Desktop/Secure/AndrewRoberts",
46    "/var/www/html/protected/TiffanyTurner",
47    "/opt/data/classified/MichellePhillips",
48    "/tmp/confidential/WilliamCampbell",
49    "/root/secret/StevenParker",
50    "/etc/private/MelissaEvans",
51    "/usr/local/hidden/AnthonyEdwards",
52    "/content/drive/MyDrive/PrivateClient/V0.7/data/MeganCollins",
53    "/MyDownloads/BusinessPlans/JohnMurphy",
54    "/home/jessica/Documents/Sensitive/JessicaMorgan",
55    "/Users/matthew/Desktop/Private/MatthewCook",
56    "/var/www/html/restricted/AshleyRogers",
57    "/opt/data/secure/KimberlyCooper",
58    "/tmp/private/JasonReed",
```

```
59   "/root/confidential/MatthewBailey",
60   "/etc/hidden/DavidRichardson",
61   "/usr/local/secret/BrianCox",
62   "/content/drive/MyDrive/PrivateClient/V0.8/data/AmandaHoward",
63   "/MyDownloads/ResearchData/DanielWard",
64   "/home/sarah/Documents/Classified/SarahPeterson",
65   "/Users/joseph/Desktop/Protected/JosephGray",
66   "/var/www/html/private/AshleyPrice",
67   "/opt/data/sensitive/KimberlyRamirez",
68   "/tmp/secure/JasonJames",
69   "/root/private/MatthewWatson",
70   "/etc/secret/DavidBrooks",
71   "/usr/local/hidden/BrianKelly",
72   "/content/drive/MyDrive/PrivateClient/V0.9/data/EmilyBennett",
73   "/MyDownloads/PersonalPhotos/DanielSanders",
74   "/home/elizabeth/Documents/Confidential/ElizabethHughes",
75   "/Users/andrew/Desktop/Secure/AndrewLong",
76   "/var/www/html/protected/TiffanyFoster",
77   "/opt/data/classified/MichelleGonzales",
78   "/tmp/confidential/WilliamBryant",
79   "/root/secret/StevenAlexander",
80   "/etc/private/MelissaRussell",
81   "/usr/local/hidden/AnthonyGriffin",
82   "/content/drive/MyDrive/PrivateClient/V1.0/data/MeganDiaz",
83   "/MyDownloads/MedicalImages/JohnHayes",
84   "/home/jessica/Documents/Sensitive/JessicaMyers",
85   "/Users/matthew/Desktop/Private/MatthewFord",
86   "/var/www/html/restricted/AshleyHamilton",
87   "/opt/data/secure/KimberlyGraham",
88   "/tmp/private/JasonSullivan",
89   "/root/confidential/MatthewWallace",
90   "/etc/hidden/DavidWoods",
91   "/usr/local/secret/BrianCole",
92   "/User/SarahMiller/Emails/To/JohnSmith",
93   "/User/DavidBrown/Emails/Drafts",
94   "/User/MeganCollins/Documents/Research/ProjectX",
95   "/User/JohnMurphy/Documents/Financial/Taxes2023",
96   "/User/JessicaMorgan/Documents/Personal/Journal",
97   "/User/MatthewCook/Documents/Work/Presentations",
98   "/User/AshleyRogers/Documents/School/Assignments",
99   "/User/KimberlyCooper/Documents/Travel/Itinerary",
100  "/User/JasonReed/Documents/Health/Records"
```

Listing 7: Full list of (fictitious) sensitive path links we use to create SD.Links.

