# OpenReview forum: "Measuring memorization in RLHF for code completion"
_ICLR.cc/2025/Conference — ICLR 2025 Poster_

### Official Review · Reviewer_v9B4 · 2024-10-28

**Soundness:** 3
**Presentation:** 2
**Contribution:** 3
**Rating:** 5
**Confidence:** 2

**Summary:**

This paper investigates the impact of reinforcement learning with human feedback (RLHF) and other preference alignment methods, such as Direct Preference Optimization (DPO) and ΨPO, on data memorization in large language models, specifically in code completion tasks. While previous research has explored data memorization during fine-tuning, how memorization behaves in the RLHF alignment process remains less understood. The study aims to shed light on the potential privacy risks of RLHF and direct preference learning methods, as sensitive user data could be inadvertently memorized and reproduced by the model. The authors find that RLHF reduces the likelihood of memorizing sensitive data from the reward modeling and RL stages compared to fine-tuning alone. However, if data is already memorized during fine-tuning, it is likely to persist post-RLHF. Conversely, direct preference learning methods like Identity Preference Optimization (IPO), a specific ΨPO method, increase the chances of data regurgitation. The study concludes that RLHF is safer for mitigating the risk of exposing sensitive data, and the findings are consistent across multiple datasets, tasks, and model scales.

**Strengths:**

1. The paper addresses a significant and underexplored aspect of data privacy in RLHF and direct preference alignment, especially concerning large language models widely deployed in code completion, a critical real-world application.
2. By comparing RLHF with direct preference learning methods like IPO, the paper provides actionable insights into safer alignment practices, offering practical relevance for model developers.
3. The study's conclusions are derived from experiments across diverse code completion datasets, tasks, and model scales, supporting the robustness and generalizability of the findings.

**Weaknesses:**

1. Wrong template.
2. It's hard to understand the whole workflow because the paper lacks a workflow figure

**Questions:**

See above

---

> ### Author Response · Authors · 2024-11-18
> **Response**
>
> Thank you for your feedback. We have included an updated revision pdf (with changes in red to make it clear what has been added).
>
> > Wrong template.
>
> Thank you for your comment. We have double checked and believe we have used the right template (ICLR 2025 style file). Could you please clarify the issue you have found?
>
> > It's hard to understand the whole workflow because the paper lacks a workflow figure
>
> Great suggestion. We have added figures describing how synthetic data is used during the course of training and how RLHF differs from IPO in Appendix A.3 (which we can promote to the main paper if the reviewer agrees they are useful). Please take a look and let us know if we could make these clearer.

---

> ### Author Response · Authors · 2024-11-25
>
> Hi reviewer v9B4,
>
> As the discussion period closes tomorrow, we wanted to draw your attention to our response addressing the feedback in your initial review. Specifically, we have verified that we use the appropriate template, and we have added a workflow figure on your recommendation. Thank you for your reconsideration of our submission.

---

### Official Review · Reviewer_1mFn · 2024-11-02

**Soundness:** 3
**Presentation:** 3
**Contribution:** 3
**Rating:** 6
**Confidence:** 1

**Summary:**

The paper analyzed in code completion models how training data memorization can surface and propagate through each phase of RLHF and direct preference learning. The findings suggests RLHF, as opposed to direct preference learning, is a safer way to mitigate the risk of regurgitating sensitive preference data when aligning large language models.

**Strengths:**

1. The paper studied a very practical and critical problem; the memorization of RLHF and its risk of data leakage.
2. The experiments are detailed.

**Weaknesses:**

I am new to RLHF with code completion model. I didn't find apparent weakness.

**Questions:**

Is there any reason of choosing code completion task for studying the RLHF memorization? Does the conclusion still hold for other RLHF scenarios? if not what else conditions should we consider?

---

> ### Author Response · Authors · 2024-11-18
> **Response**
>
> Thank you for your feedback and for recognizing the practical importance of our study on memorization risks in RLHF. We appreciate your supportive comments regarding the experiment detail and significance of the problem. We have included an updated revision pdf (with changes in red to make it clear what has been added).
>
> > Is there any reason of choosing code completion task for studying the RLHF memorization? Does the conclusion still hold for other RLHF scenarios? if not what else conditions should we consider?
>
> We choose code completion as it is one of the most commonly deployed use cases of a model trained via RLHF (in addition to conversational AI assistants). We believe our results should translate to other RLHF scenarios, but it would be interesting future work to measure memorization of data in other scenarios, e.g. conversational AI. We do note however that our results hold over natural language datasets (see section 5.5).

---

### Official Review · Reviewer_Xfcj · 2024-11-03

**Soundness:** 4
**Presentation:** 4
**Contribution:** 3
**Rating:** 8
**Confidence:** 4

**Summary:**

This paper presents the first comprehensive study of memorization in Reinforcement Learning from Human Feedback (RLHF) pipelines, focusing on code completion as a practical use case. The authors analyze how training data memorization manifests across different stages of RLHF (fine-tuning, reward modeling, and RL fine-tuning) and compare it to direct preference learning approaches like IPO.

Using a synthetic dataset and robust evaluation methodology, they find that RLHF significantly reduces memorization of reward model training data compared to direct fine-tuning, while direct preference learning exhibits higher memorization rates. The results are validated across different model scales and domains.

This paper makes important contributions to our understanding of memorization in RLHF pipelines, with clear practical implications. The methodology is sound and the findings are well-supported. While there are some limitations, they do not significantly detract from the paper's contribution.

**Strengths:**

1. Thorough experimental methodology with appropriate controls and metrics
2. Important practical implications for deploying large language models
3. Novel and significant findings about RLHF's memorization properties
4. Validation across multiple scales, domains, and datasets
5. Clear writing and comprehensive presentation of results
6. Strong technical foundation and careful experimental design

**Weaknesses:**

1. Main experiments focus on one synthetic dataset, though results are validated on other datasets
2. Could explore a wider range of model architectures and scales
3. Some hyperparameter choices could be better justified
4. Analysis could include more direct preference learning methods beyond IPO
5. Could provide more detailed analysis of failure cases and limitations

**Questions:**

1. How sensitive are the results to the choice of synthetic dataset construction? Would different types of sensitive information show different memorization patterns?
2. How do the findings generalize to other direct preference learning methods beyond IPO?
3. What specific recommendations would you make to practitioners implementing RLHF pipelines based on these findings?
4. How do the memorization patterns change with extremely large model scales (e.g., >100B parameters)?
5. How do you think your results might generalize to other common applications of RLHF, such as conversational AI, where memorization of personal or sensitive information could be equally problematic?
6. Are there any scenarios where learning directly through human preferences (like IPO or DPO) might be a better tradeoff despite the increased likelihood of memorization?

---

> ### Author Response · Authors · 2024-11-18
> **Response 1 of 2**
>
> Thank you for your detailed feedback and appreciation of our work. We value your constructive suggestions, which we believe will enhance the clarity and impact of our paper. Below, we address each of your comments, and have included an updated revision pdf (with changes in red to make it clear what has been added).
>
> > Main experiments focus on one synthetic dataset, though results are validated on other datasets
>
> Thank you for your feedback. Our study utilizes two distinct synthetic datasets, SD.Base and SD.Links, to capture different memorization patterns. SD.Base includes examples of cryptographic tasks that allow us to measure full example memorization, simulating proprietary or institutional knowledge. SD.Links, on the other hand, consists of PII-like file paths, enabling us to measure memorization of privacy-sensitive information by assessing whether the model regurgitates these paths. We selected these datasets to closely examine memorization risks in code completion contexts, especially where privacy is a concern.
>
> > Could explore a wider range of model architectures and scales
>
> Thank you for this suggestion. In addition to Gemini Nano-1 (1.8B), we tested memorization behavior across several model scales, including Gemma 2B, Gemma 7B, Gemini Pro, and T5-Base. These experiments consistently showed similar memorization patterns across model scales, supporting the generalizability of our findings. We will highlight these results in the manuscript to underscore the study’s cross-scale applicability.
>
> > Some hyperparameter choices could be better justified
>
> Thank you for pointing this out. We detail our hyperparameter choices in Table 2 in Appendix I and these hyperparameters are selected via grid search on validation data during training. We have added an explanation of how we choose hyperparameters in appendix A.3.
>
> > Analysis could include more direct preference learning methods beyond IPO
>
> We agree that it would be useful to include preference learning methods beyond IPO. Direct Preference Optimisation (DPO), another popular preference learning algorithm, is a special case of the general algorithm \Psi PO (of which IPO is also a special case). In contrast to DPO, IPO includes an extra regularization term to further prevent overfitting to preference data. Since we observe memorization when training with additional regularization in IPO, we would expect our results to extrapolate to DPO where the lack of regularization will allow the model to more easily overfit and memorize preference data.  For this reason, we believe our results should extrapolate to other preference learning methods.
>
> > Could provide more detailed analysis of failure cases and limitations
>
> Thank you for your comment. We include analysis on differences in memorization rates between the 'base' and 'links' datasets in the paper, and would appreciate your recommendation on additional failure analysis we could include to strengthen our paper.
>
> > How sensitive are the results to the choice of synthetic dataset construction? Would different types of sensitive information show different memorization patterns?
>
> Our synthetic dataset was intentionally constructed with various types of sensitive data, such as file paths and cryptographic data, to measure how memorization risks vary by data type. We have observed that variations in dataset composition, particularly involving different types of sensitive data, can lead to slight differences in memorization patterns. For instance, in section 5.1, we observe that the data in SD.Links (PII data in link format) exhibits higher memorization rates than the data in SD.Base (code containing sensitive information like library versions).
>
> > How do the findings generalize to other direct preference learning methods beyond IPO?
>
> We believe our findings generalize well to other preference learning methods as IPO and DPO are specific instances of a more general preference learning algorithm (\Psi PO). Moreover, the main difference between IPO and DPO is that IPO introduces a regularization term to prevent overfitting compared to DPO, which suggests that if we observe memorization when training with IPO, we should expect memorization when training with DPO (as the lack of regularization term should allow the model to more easily overfit and memorize).
>
> > What specific recommendations would you make to practitioners implementing RLHF pipelines based on these findings?
>
> Thank you for your question. Based on our findings, we recommend using RLHF over direct preference learning methods when dealing with potentially sensitive data due to RLHF’s lower memorization tendency. Additionally, we advise tuning the KL regularization coefficient carefully during RL fine-tuning to balance memorization risks with model fidelity, as well as conducting regular model audits for sensitive data leakage. These recommendations have been added to the discussion.

---

> > ### Author Response · Authors · 2024-11-18
> > **Response 2 of 2**
> >
> > > How do the memorization patterns change with extremely large model scales (e.g., >100B parameters)?
> >
> > Thank you for your question. We agree this is an interesting question for future work – our work, and prior work, suggest that larger models have higher capacity for memorization in the finetuning and pretraining setting, and it would be interesting to see whether/how this transfers to RLHF. We have not observed a substantial increase in memorization in RLHF with increased model capacity, so our hypothesis is that memorization risk should still be reduced in extremely large model scales (but this is partly speculation).
> >
> > > How do you think your results might generalize to other common applications of RLHF, such as conversational AI, where memorization of personal or sensitive information could be equally problematic?
> >
> > Our findings have direct implications for conversational AI, where RLHF is used extensively and memorization of sensitive user data poses a risk. We believe that RLHF’s regularization mechanisms could mitigate memorization similarly in dialog models, making it a practical choice for privacy-conscious applications. We note that we also have results on natural language datasets such as LIMA (Zhou et al., 2024) and Anthropic HH (Bai et al., 2022), see section 5.5.
> >
> > > Are there any scenarios where learning directly through human preferences (like IPO or DPO) might be a better tradeoff despite the increased likelihood of memorization?
> >
> > This is a great question. We believe the tradeoff is primarily the computational burden of executing a full RLHF pipeline (more complex) versus direct preference learning (simpler, but higher memorization risk). We’ve added a sentence to the discussion that comments on this in addition to recommendations for practitioners.

---

> > > ### Comment · Reviewer_Xfcj · 2024-11-24
> > > **Response to authors**
> > >
> > > Thank you for your clarifications. I have increased my score accordingly.

---

### Official Review · Reviewer_NUep · 2024-11-06

**Soundness:** 2
**Presentation:** 3
**Contribution:** 3
**Rating:** 6
**Confidence:** 3

**Summary:**

The paper discovers that RLHF significantly decreases the chance that data used for reward modeling and reinforcement learning is memorized in comparison to directly fine-tuning on this data, but that examples already memorized during the fine-tuning stage of RLHF, will, in the majority of cases, remain memorized after RLHF. The paper finds that aligning by learning directly from human preference data via Identity Preference Optimization (IPO) increases the likelihood that training data is regurgitated compared to RLHF. The work suggests that RLHF, as opposed to direct preference learning, is a safer way to mitigate the risk of regurgitating sensitive preference data when aligning large language models.

**Strengths:**

(1) The observations in this paper are interesting.

(2) The observations can potentially help important real-world applications.

(3) The paper is well written and easy to understand.

**Weaknesses:**

(1) It would be interesting to show the observations in this paper are generally applicable in various model backbones with various sizes. It seems that currently only Gemini Nano-1(1.8B) is used as the base mode in the experiments.

(2) Related to the statement that IPO exhibits stronger memorization of preference data than RLHF, currently the result is obtained using one model backbone. It would be great to validate this statement using various model backbones. Besides, it seems that this statement is only validated by some empirical results. It would be great if the paper can provide some in-depth (theoretical) analysis to illustrate why IPO exhibits stronger memorization of preference data than RLHF.

(3) Related to the datasets and reward model design used in this paper, the paper may provide more discussions to explain how they are chosen, and how they are similar to (or different from) the choices of the datasets and reward model design used in previous works related to RLHF for code completion (or similar tasks).

**Questions:**

(1) Could you please provide some in-depth (theoretical) analysis to illustrate why IPO exhibits stronger memorization of preference data than RLHF?

(2) Related to the datasets and reward model design used in this paper, the paper may provide more discussions to explain how they are chosen, and how they are similar to (or different from) the choices of the datasets and reward model design used in previous works related to RLHF for code completion (or similar tasks).

---

> ### Author Response · Authors · 2024-11-18
> **Response**
>
> Thank you for your constructive feedback and insightful suggestions. We appreciate your recognition of the paper's potential impact and readability. Below, we address each of your comments, and have included an updated revision pdf (with changes in red to make it clear what has been added).
>
> > (1) It would be interesting to show the observations in this paper are generally applicable in various model backbones with various sizes. It seems that currently only Gemini Nano-1(1.8B) is used as the base mode in the experiments.
>
> We agree that exploring additional model backbones and sizes enhances the generalizability of our findings. While Gemini Nano-1 (1.8B) was a focal point in our study, we also increased the scale by including experiments with Gemma 2B, Gemma 7B, Gemini Pro, and T5 Base models (section 5.4 and 5.5). These expanded experiments show consistent patterns in memorization behaviors, reinforcing the applicability of our observations across different scales and architectures. We hope these results address your concerns regarding additional model backbones and sizes.
>
> > (2) Related to the statement that IPO exhibits stronger memorization of preference data than RLHF, currently the result is obtained using one model backbone. It would be great to validate this statement using various model backbones. Besides, it seems that this statement is only validated by some empirical results. It would be great if the paper can provide some in-depth (theoretical) analysis to illustrate why IPO exhibits stronger memorization of preference data than RLHF.
>
> We agree that the empirical results showing IPO exhibits stronger memorization characteristics would be well complemented by theoretical motivation illustrating why direct preference learning might be expected to exhibit such behavior. As memorization is a nascent field and almost all prior work on memorization we are aware of is empirical, we believe this is a compelling direction for future work.
>
> > 3) Related to the datasets and reward model design used in this paper, the paper may provide more discussions to explain how they are chosen, and how they are similar to (or different from) the choices of the datasets and reward model design used in previous works related to RLHF for code completion (or similar tasks).
>
> We appreciate your suggestion for a more thorough discussion of our dataset and reward model choices. Our synthetic dataset (SD) was designed to include sensitive information (e.g., PII-like paths, cryptographic keys) to rigorously assess memorization in a high-stakes context. While we are unaware of existing work studying memorization in code completion models trained via RLHF, we have amended the manuscript to discuss how our datasets align with previous works studying RLHF models for code completion (which don’t focus on memorization) to show how our datasets align with how code completion models are standardly trained. Please see new details in Appendix A.1.

---

### Author Response · Authors · 2024-11-22

Dear reviewers,

As the discussion period draws to a close, we'd like to check in on whether you've had a chance to review our responses and have any follow up questions. We appreciate your feedback, thank you in advance!

---

### Meta-Review · Area_Chair_Sf8Z · 2024-12-20

**Metareview:**

This paper analyzes the effects of training data memorization in RLHF and direct preference learning, particularly in the context of code completion models. It concludes that while RLHF reduces the likelihood of memorization compared to direct fine-tuning, memorized examples from the fine-tuning stage persist after RLHF; conversely, aligning through IPO tends to increase the risk of regurgitating training data, suggesting RLHF is a safer approach for mitigating this risk. The paper is well-written, and the authors address almost all of the reviewers' concerns.

**Additional Comments On Reviewer Discussion:**

The paper is well-written, and the authors address almost all of the reviewers' concerns.

---

### Decision · Program_Chairs · 2025-01-22

Accept (Poster)